# Physical Activity and Sedentary Patterns of Pregnant Women in Southern Spain and the Relationship with Sociodemographic and Obstetric Characteristics: A Cross-Sectional Study

**DOI:** 10.3390/healthcare13121423

**Published:** 2025-06-13

**Authors:** Katty M. Cavero, Rita Santos-Rocha, Diego Gómez-Baya, Silvia Rosado-Bello, Elia Fernández-Martínez, Mónica Maure-Rico, Anna Jean Grasmeijer, Ramón Mendoza-Berjano

**Affiliations:** 1Doctorate Programme on Health Sciences, University of Huelva, 21007 Huelva, Spain; katty.cavero@gmail.com; 2ESDRM, Department of Physical Activity and Health, Sport Sciences School of Rio Maior, Santarém Polytechnic University, 2040-413 Rio Maior, Portugal; 3SPRINT, Sport Physical Activity and Health Research & Innovation Center, 2040-413 Rio Maior, Portugal; 4Research Group on Health Promotion and Development of Lifestyle across the Life Span, University of Huelva, 21007 Huelva, Spain; diego.gomez@dpee.uhu.es (D.G.-B.); anna-jean.grasmeijer983@alu.uhu.es (A.J.G.); ramon@uhu.es (R.M.-B.); 5Department of Social, Developmental and Educational Psychology, University of Huelva, 21007 Huelva, Spain; 6Obstetrics Unit, Department of Obstetrics and Gynaecology, Juan Ramon Jimenez University Hospital, 21005 Huelva, Spain; silvia_rosabello@live.com; 7Department of Nursing, University of Huelva, 21071 Huelva, Spain; elia.fernandez@denf.uhu.es; 8Mallén Health Centre, Andalusian Health Service, 41071 Seville, Spain; monica.maure.sspa@juntadeandalucia.es

**Keywords:** physical activity, sedentary behaviour, pregnancy, patterns, correlates, guidelines, Spain

## Abstract

**Background/Objectives**: Physical activity (PA) during pregnancy presents health benefits for mother and child. The World Health Organization (WHO) recommends at least 150 min of moderate-intensity physical activity per week for a healthy pregnancy. The objectives of this study were to describe physical and sedentary activity patterns, estimate the proportion of women meeting PA recommendations, and identify associated sociodemographic and obstetric characteristics in a sample of pregnant women from southern Spain. **Methods**: For this cross-sectional study, a random sample of 385 pregnant women attending their 20th-week scheduled ultrasound at their referral hospital was selected. Inclusion criteria were being between 18 and 22 gestational weeks pregnant and communicating in Spanish. A face-to-face structured interview was used to collect demographic, obstetric, and PA data, as well as sedentary patterns. Indexes of weekly PA in various domains were computed. Bivariate analyses were conducted to assess the variability of physical and sedentary activities according to sociodemographic and obstetric variables. Statistical significance was set at *p* < 0.05. **Results**: A total of 84.4% of participants engaged in some type of PA and 73.7% met WHO PA requirements. Higher leisure PA was associated with higher education (*p* < 0.05) and first-time pregnancy (*p* < 0.01). Higher work PA was linked to lower education (*p* < 0.01), being born outside Spain (*p* < 0.05), and later pregnancy awareness (*p* < 0.01). Higher sedentary daily time was associated with higher education (*p* < 0.01), speaking Spanish as a child (*p* < 0.05), and first-time pregnancy (*p* < 0.05). **Conclusions**: Most pregnant women in this study met PA recommendations. Correlates of leisure PA differ from those regarding work PA.

## 1. Introduction

Engaging in physical activity (PA) during pregnancy is beneficial for the health of mother and offspring [1]. Health advantages for the mother include less risk of developing obesity, cardiovascular diseases, hypertensive disorders, and gestational diabetes mellitus [2,3,4]. Maternal PA also decreases musculoskeletal discomfort [5], prevents excessive gestational weight gain and postpartum weight retention [6], and is associated with lower levels of systemic inflammation [7]. With regard to mental health, PA during pregnancy improves psychological well-being by reducing anxiety, stress, and prenatal depression [8,9,10,11] and lowers the risk of postpartum depression [12].

Regarding advantages for the baby, PA during pregnancy does not have negative effects on fetoplacental development and is not associated with neonatal complications [13]. On the contrary, it is linked to advanced neurobehavioral maturation, healthier foetal growth, improved stress tolerance in the baby [2,3,14], and lowered neonatal adiposity [15]. It also has a positive long-term impact on the body composition and health outcomes of the child [13,15]. Engaging in PA during pregnancy positively influences lung development in the foetus, lung functioning in the newborn, and lowers the risk of developing asthma [16].

Besides the health benefits for mother and child, maternal PA lowers the risk of complications at delivery, such as preterm births for women without pregnancy complications [17], c-section births, and vulvo-vaginal tears [10,14,18]. Moreover, PA and exercise during pregnancy are also associated with less frequent premature rupture of membranes [19], reduce potential complications for women at risk of preterm delivery [20], and tend to reduce the length of delivery and postpartum recovery [5].

Given the benefits of engaging in PA during pregnancy, various organisations interested in the well-being of women during and after pregnancy as well as the health of the offspring have updated recommendations on this topic. The World Health Organization (WHO) recommends performing at least 150 min of moderate-intensity aerobic PA per week for a healthy pregnancy [1]. Similar guidelines have been provided around the world; in fact, recommendations of international public health guidelines on PA during gestation are similar across organisations [21].

Achieving a global perspective on PA patterns during pregnancy across world regions is not an easy task due to the paucity of studies in low-income countries and to the diversity of methods used for measuring PA. Nevertheless, there are indications of relevant differences in this regard among countries [22]. There are also some similarities in the prevalence of PA among samples of pregnant women from different countries. For instance, a study conducted in the United States on recreational PA with women in early pregnancy reported that 61% of women participated in regular PA [23]. The same figure (a prevalence of 61%) was found in a study carried out in Spain, which focused on moderate PA (MPA) among pregnant women in the second trimester [24].

In terms of meeting the stipulated recommendations of PA during pregnancy, studies have also obtained mixed results. Various researchers have reported high compliance with recommendations for PA during pregnancy. For instance, a study conducted in the city of Donostia-San Sebastian, in the north of Spain [25], collected data on PA related to housework, walking, biking, and sports during the first and second trimesters. It concluded that, overall, 80% of pregnant women met recommendations from international guidelines such as the United States Department of Health and Human Services and WHO [26,27]. Similarly, a study on leisure time PA in women in early pregnancy in Serbia concluded that about 73% of pregnant women met WHO PA recommendations [28]. In a study carried out in Sweden, almost half of the sample (47.1%) performed the recommended level of PA [29]. In the United States, a study on pregnant Hispanic women and PA related to household chores, caregiving, transport, and work found that about 45% of participants in early pregnancy met the recommended levels of PA based on ACOG guidelines [30].

However, low compliance with recommendations has also been observed. For example, a systematic review of PA among pregnant Chinese women estimated the average prevalence of meeting recommended levels of PA at 21% [31]. In a sample of pregnant women in Naples, Italy, only 37% reported performing at least 150 min of moderate intensity physical activity per week [32]. In Canada, only 27.5% of pregnant women met the PA levels recommended by the Canadian Physical Activity Guidelines [33]. Similarly, in South Africa, a cross-sectional study in Eastern Cape Province collected data on pregnant women and PA related to household chores, caregiving, occupation, and leisure time and found that approximately 26% of pregnant women met the recommended levels of PA based on international guidelines [34]. In addition, a study in Brazil about pregnancy and PA related to domestic and caregiving work, labour, and leisure time found very low compliance with international recommended levels of PA during the first (7.2%), second (7.6%), and third (4.7%) trimesters [35].

Furthermore, associations between maternal PA and demographic and obstetric characteristics have been observed. Researchers have found that, among pregnant women, higher education is associated with higher levels of leisure PA [23,28,34,36,37,38,39,40] as well as meeting PA recommendations for pregnant women [36,40,41]. Nevertheless, the relationship between different domains of PA apart from leisure time PA and educational level among pregnant women has been scarcely studied.

Several studies have explored the relationship between gravidity and PA during pregnancy, with controversial findings. In some studies, multigravidity is associated with more frequent PA during pregnancy [25,30], whereas other studies have found an association between primigravidity and a higher level of PA among pregnant women [23,35,39].

Regarding sedentary behaviour during pregnancy, the WHO warns that greater amounts of sedentary behaviour are linked to poor health outcomes, such as a higher risk of cardiovascular diseases, cancer, or type 2 diabetes. It recommends that pregnant women should limit the amount of time spent being sedentary and replace sedentary behaviour with more physical activity of any intensity [1]. Others have observed the following: excessive sitting time during the first and second trimesters of pregnancy can negatively affect placental weight [42]; moreover, sedentary behaviour is associated with an increased risk of preterm births, admissions to the neonatal intensive care unit [43], poor sleep quality [44], and prenatal and postnatal depression [45]. Longitudinal studies addressing sedentary behaviour during pregnancy have found that almost all pregnant women decreased PA levels and increased their sedentary behaviour [15,46]. Lynch et al. [30] highlighted those younger and more educated women decreased their sedentary behaviour compared to older and less educated women. Unfortunately, compared to PA, sedentary behavioural patterns during pregnancy have been scarcely studied [33,47,48].

Exploring the prevalence and understanding the correlates of PA and sedentary behavioural patterns in this population supports the planning and implementation of effective public health strategies, including suitable communication plans, to foster an active and healthy lifestyle during pregnancy.

Considering the current mixed results in the literature on physical and sedentary activity behaviours among pregnant women, the objectives of this study were threefold: (1) to describe physical and sedentary activity patterns among pregnant women in southern Spain; (2) to estimate the proportion of pregnant women meeting the recommended levels of PA according to the WHO, and; (3) to assess sociodemographic and obstetric variability regarding physical and sedentary activities of pregnant women.

## 2. Materials and Methods

### 2.1. Study Design

A cross-sectional study was conducted with a representative sample of pregnant women in a health district.

### 2.2. Setting

Recruitment of participants took place in the waiting room of the outpatient clinics of their referral public (university) hospital in Huelva, a city in southern Spain. Pregnant women, attending their 20th-week scheduled ultrasound visit, were invited to participate sequentially by a trained health professional. Detailed study information was provided, and their voluntary participation was requested. In the case that an invited pregnant woman declined participation, the next eligible pregnant woman was approached. Those who agreed to participate were guided to a private area within the outpatient clinic to sign the informed consent form. Following consent, a face-to-face interview was conducted using a structured questionnaire with all collected data anonymized. For underaged participants (<18 years of age), parental consent was obtained. Data collection was performed from April to July 2021.

### 2.3. Participants

Inclusion criteria included being between 18 and 22 gestational weeks pregnant, attending a scheduled ultrasound in the 20th week of gestation at an outpatient clinic of the referral hospital, and being able to communicate in Spanish. No criteria for age were established.

A total of 829 women attended the hospital outpatient clinic. Due to language barriers (i.e., not being fluent in Spanish) and the lack of availability of interpreters for assisting interviews, 134 of them were not invited to participate in this study. Another 174 women were not invited because the researcher was occupied at the time interviewing another pregnant woman. There was no other reason why these pregnant women were not invited to participate in this study, other than the fact that the researcher was interviewing another woman. In this regard, the sample was randomly selected. Finally, a total of 521 women were invited to participate in this study.

### 2.4. Variables

Demographics, obstetrics, physical activity, and sedentary behavioural data during pregnancy were collected. Sociodemographic data included age; educational level, grouped into four categories: no studies, compulsory studies (typically ages 6–16), pre-university studies (typically ages 17–19), and higher education (university level and beyond); currently studying (yes/no); employment status, grouped into four categories: full time, part time, unemployed, or other; population size of the place of residence; relationship status; language spoken at home currently and during childhood; and country of birth.

Obstetric variables included gravidity (including previous miscarriages, abortions, and the current pregnancy), number of vaginal and caesarean births (excluding abortions), number of miscarriages, number of abortions, health problems during pregnancies ended in live birth, age at first pregnancy, high-risk pregnancy consultation, pregnancy planning, assisted reproduction pregnancy, trimester of pregnancy awareness (first/second), professional health care during pregnancy, date of initiation of professional control of current pregnancy, date of last menstrual period, and self-reported height and weight to determine body mass index (BMI).

For physical activity and sedentary behaviour, 16 variables were collected through the Global Physical Activity Questionnaire, Version 2 (GPAQv2) [49]. See the questionary description in the measurements section below for more details.

### 2.5. Data Measurments

Demographic and obstetric data related to pregnancy were collected using a series of multiple-choice questions included in a questionnaire developed by the research team in a previous study [50,51]. Some questions required verbatim answers, which were categorised after the interview by the research team.

PA data was collected using the GPAQv2 which was tested in nine countries between 2002 and 2005 to collect physical activity information relevant to both developed and developing countries. Its validity and reliability have been demonstrated to be appropriate to measure PA in adult populations [52]. A subsequent study provided low-to-moderate validity and acceptable reliability evidence for the GPAQ [53]. In addition, the GPAQ yielded consistently reliable results, good-to-very good test–retest reliability across intervals of three days to two weeks [54].

The GPAQ consists of 16 questions regarding physical activity at work, travel to and from places, recreational activities, and sedentary behaviour. In the “activity at work” domain, the GPAQ refers to a wide range of possibilities (e.g., paid or unpaid work, study/training, domestic and carving tasks, harvesting food/crops, fishing or hunting for food, and seeking employment). To collect physical activity information in these domains, participants were asked about the frequency (i.e., number of days) of the following activities in a typical week and their duration (i.e., amount of time in hours and minutes) on a typical day: (1) vigorous work PA, (2) moderate work PA, (3) active travel to and from places, (4) vigorous recreational PA, (5) moderate recreational PA, and (6) sitting/resting time. The protocol for collecting data using the GPAQ states that PA must be performed for at least 10 min continuously to be recorded. Sedentary behaviour was assessed by a single question, namely, “How much time do you usually spend sitting or reclining on a typical day?”.

Furthermore, following recommendations of the GPAQ protocol, visual aids were used during the interviews in the form of cards that represented a wide range of vigorous and moderate activities related to the workplace, and recreational and sports activities. These visual aids had been adapted and piloted prior to the beginning of this study to represent activities performed by women in the local context where this study was to be conducted.

### 2.6. Bias

Efforts were made throughout this study design and execution to minimise potential sources of bias. To mitigate selection bias, a clear and consistent recruitment protocol was implemented to enrol eligible participants systematically from the outpatient clinics, aiming for a representative sample within the study population. Participation was voluntary, and anonymity was ensured to encourage honest responses. Research personnel were thoroughly trained to ensure consistent data administration and recording. While acknowledging the potential for recall bias inherent in self-reported data, clear instructions and reference periods were provided to participants to aid the accuracy of data collected. Furthermore, potential confounding factors, such as age, education level, and parity were collected and used in statistical analyses.

### 2.7. Study Size

The study size was primarily determined by practical considerations, logistical feasibility, and the objective to provide a robust and detailed description of physical activity and sedentary patterns within our target population. A concerted effort was made to recruit over 350 eligible pregnant women from a city of southern Spain (Huelva) within the designated study period (April–July 2021). This approach aimed to ensure a comprehensive characterisation of these behaviours and their associated sociodemographic and obstetric factors, thereby contributing valuable descriptive data to the existing literature.

### 2.8. Quantitative Variables

Physical activity data were cleaned and analysed following the steps from the GPAQ analysis guide [55]. Overall, the number of missing values was less than 1% across all questions.

First, a univariate analysis was performed for each variable. Next, a series of variables were generated to calculate the Metabolic Equivalent of Task (MET) minutes. These new variables also underwent a univariate analysis. For these calculations, frequency and duration of moderate work-related PA, vigorous work-related PA, active travel to and from places, leisure time moderate PA, and leisure time vigorous PA were converted to minutes and added up for each participant to calculate the total minutes per week of each activity domain. Next, totals for moderate PA (work, travel to and from places, and leisure) and totals for vigorous PA (work and leisure) were calculated and converted to MET-minutes to obtain the total PA MET minutes per week.

The GPAQ analysis instructions were followed to transform energy expenditure into METs. When calculating a participant’s overall energy expenditure related to physical activity, 4 METs were assigned to the time spent in moderate activities and 8 METs were assigned to the time spent in vigorous activities [53,55]. Therefore, to convert physical activity time to MET minutes by week, the total time spent performing moderate activities in a typical week was multiplied by 4, and the total time spent in vigorous activities was multiplied by 8. Sedentary behaviour was only measured in minutes on a typical day.

### 2.9. Statistical Methods

Bivariate analyses to test the relationship between physical activity and sociodemographic and obstetric variables were performed applying the Chi-squared test of independence. Normality of the quantitative variables was tested using the Kolmogorov–Smirnov goodness of fit test. The ANOVA test was applied for bivariate analysis in case of normality, and the Mann–Whitney and Kruskal–Wallis tests for non-parametric data. Statistical significance was set at *p* < 0.05. Data analysis was concluded using the Software Package for Social Sciences (SPSS version 29).

### 2.10. Ethics

The Research Ethics Committee of the Andalusian Health and Families Department reviewed and approved the study protocol. Verbal and written information about this study was given to participants and informed consent forms were signed by all participants before data collection started. All data were anonymous. The 1975 Helsinki declaration and its subsequent amendments were respected.

## 3. Results

### 3.1. Sociodemographic and Obstetric Characteristics

Of the 521 women who were invited to participate in this study, 117 refused. A total of 404 pregnant women provided signed consent and were subsequently interviewed. The response rate was 77.5%. The mean age of women unwilling to participate (30.0 years) was very similar to that of the final sample (31.8). Yet, with regard to educational level, the percentage of women with higher education among those who refused participation (17.9%) was lower than in the sample (30.9%).

After data collection, 19 women were excluded from the analysis because they did not meet the gestational age criterion. A total of 385 women aged 14 to 48 years with a mean age of 31.8 (SD = 5.99) were included in the final sample. Only 4 women were under 18 years. Approximately 40% of women were over the age of 35. About 46% of the sample had primary education, while 30.9% had higher education. Less than half of the participants worked full time (41%). In addition, two thirds of the sample (67%) lived in towns with a population greater than 20,000 and around 92% of pregnant women had been born in Spain. Almost all women (99.2%) reported being in a relationship at the time of the interview. Based on the Body Mass Index, nearly half of the sample (48%) was considered within normal weight, about 30% was classified as overweight and 21% lived with obesity (Table 1).

The average number of pregnancies of participants was 2.1 (SD = 1.1) with a mean age of 27.5 years (SD = 6.4) for the first pregnancy. Almost two thirds of women (63.1%) had been pregnant before, and 16.1% of women reported having had problems during previous pregnancies that ended in live birth. Moreover, more than three quarters of women (76.6%) had planned the current pregnancy, 12.7% had used assisted reproduction for their current pregnancies, and 15.6% were receiving high-risk care (Table 2).

### 3.2. Physical and Sedentary Activity Patterns

The mean of total PA (work, travel to and from places, and leisure) was 902.9 min/week (SD = 1061.3). The mean was 596.3 min/week (SD = 1037.5) for work-related PA, 97.6 min/week (SD = 211.3) for active travel to and from places, and 212.1 min/week (SD = 262.3) for leisure time PA. On average, women spent 700.3 (SD = 858.6) min/week performing moderate-intensity PA (work, travel to and from places, and leisure) and 202.6 (SD = 681.8) min/week performing vigorous-intensity PA (work and leisure). Finally, the average time spent on sedentary behaviour on a typical day was 234.6 min (SD = 162.99) (Table 3).

Approximately 84.4% of the pregnant women in this study reported engaging in some type of PA. The proportion of women who reported engaging in PA was 33.5% as part of their work and 36.2% for travel to and from places; however, close to twice as many women (61.7%) engaged in recreational PA. Additionally, only 10.9% of women engaged in vigorous PA (work and leisure) while 81.3% engaged in moderate PA (work, travel to and from places, and leisure) (Table 3).

### 3.3. Proportion of Pregnant Women Meeting the WHO Recommendation

The main WHO physical activity recommendation for pregnant women is to perform at least 150 min of moderate intensity physical activity per week [1]. Based on this criterion, our results indicated that 73.7% of pregnant women in this study met the requirement of PA for health (Table 3).

### 3.4. Variability of Compliance with WHO Physical Activity Recommendations for Health According to Sociodemographic and Obstetric Variables

The results yielded significant variability in meeting the WHO recommendation according to one sociodemographic variable—currently studying—and one obstetric variable—gravidity. Compliance with PA recommendations was greater among women who were currently studying (86%), compared to those who were not (71.6%), χ^2^(1, N = 283) = 5.2, *p* = 0.02, V = 0.12. About 80% of first-time pregnant women reported greater compliance with PA recommendations for health compared to around 70% of multigravida women, χ^2^(1, N = 283) = 5.04, *p* = 0.02, V = 0.11. (Table 4).

### 3.5. Variability of Physical Activity by Domain, MET Minutes/Week and Sedentary Behaviour According to Sociodemographic Variables

A relevant variability in two of the three domains of PA regarding sociodemographic variables was found. First, minutes of leisure time PA varied significantly according to level of education and whether women were currently studying. Leisure time PA and educational level were positively correlated. On average, women with higher levels of education spent more minutes of recreational PA per week compared to those with lower education (Figure 1 and Appendix A). Moreover, women who were currently studying engaged in more time of leisure PA per week. Pregnant women who were studying spent an average of 292.5 min/week in leisure PA (SD = 375.01), while the mean time for women who were not studying was 198.1 min/week (SD = 235.3) (*p* < 0.05) (Appendix A).

Work-related PA also showed variability according to educational level and whether women were currently studying. Women with lower levels of education reported performing more PA as part of their work than women with higher education. Specifically, women with compulsory schooling had the highest mean for work-related PA minutes per week (M = 916.4, SD = 1233.3) (Figure 2 and Appendix A). Regarding the “currently studying” variable, work-related PA also showed relevant variability. Women who were not currently studying tended to engage in more than twice the amount of weekly time of work-related PA (M = 658.1, SD = 1085.1) compared to women who were studying (M = 244.2, SD = 602.1; *p* < 0.01). In addition, work-related PA was also significantly related to country of birth. Women who were born in Spain engaged in lower amounts of work-related PA minutes per week (M = 568.8, SD = 1027.8) compared to women born elsewhere (M = 897.2, SD = 1110.96; *p* < 0.05) (Figure 3 and Appendix A). No significant variability was found between travel to and from places PA and any sociodemographic characteristics (Appendix A).

When PA patterns were measured in terms of MET-min/week, a relevant variability was found regarding two sociodemographic variables: educational level and population of the place of residence. Among the four educational level groups, women with lower levels of education performed approximately twice as many MET-min/week than women with higher levels of education. In fact, women with compulsory schooling had the highest average MET-min/week (M = 6294.8, SD = 7813.6) (Figure 4 and Appendix A). Regarding the size of the place of residence, the smaller the town, the greater the MET-min/week. Specifically, women who lived in a town with a population greater than 50,000 people spent an average of 3655.8 MET-min/week (SD = 6004.3) compared to 5446.7 MET-min/week (SD = 8075.1) in those who lived in a town with a population under 5000 (*p* < 0.05) (Appendix A).

Daily time of sedentary behaviour exhibited significant variability in relation to educational level and language spoken at home as a child. Educational level and time spent in sedentary behaviour were positively correlated. Specifically, women with no studies spent an average of 105 min/day in sedentary behaviour (SD = 51.96), whereas women with higher education engaged in an average of 281 min/day of sedentary time (SD = 192.1; *p* < 0.01) (Figure 5). Regarding language spoken at home as a child, women who spoke Spanish while growing up reported a higher average of daily sedentary time (M = 239.3, SD = 165) compared to women who spoke another language (M = 189.6, SD = 136.3) (*p* < 0.05) (Appendix A).

### 3.6. Physical Activity by Domain, MET-Minutes, and Sedentary Behaviour Variability in Relation to Obstetric Variables

Regarding the relationship between the three physical activity domains and the obstetric variables, results showed significant variability in two domains: leisure time PA according to gravidity, and work-related PA based on trimester of pregnancy awareness. On average, primigravida women spent more min/week (254.3; SD = 242.7) performing leisure time PA than women with previous pregnancies (M = 187.3, SD = 270.6) (*p* < 0.01) (Appendix A). In addition, a great difference was detected in average time of work PA according to trimester of pregnancy awareness. Women who became aware of their pregnancy during the second trimester reported about four times more work-related PA (M = 2124, SD = 741.4) than those who found out during the first trimester (M = 569.9, SD = 1020.8) (*p* < 0.01) (Appendix A).

Regarding MET-min/week, there was no significant variability with obstetric variables except for one: trimester of pregnancy awareness. The earlier women found out about their pregnancy, the fewer MET-min/week they performed. The average METs for women who learned about their pregnancy during the first trimester was 4273.6 (SD = 6154.5). In women who learnt the news of their pregnancy during the second trimester, the average was 13,836 MET-min/week (SD = 6577.5) (*p* < 0.01) (Appendix A).

Finally, a relevant variability in time of sedentary behaviour per day was observed according to gravidity. First-time pregnant women spent more time involved in sedentary behaviours than women who had had more than one pregnancy. Specifically, primigravida women engaged in approximately 252 min/day of sedentary behaviour (SD = 160.4) compared to women who had been pregnant before, who spent about 225 min/day being sedentary (SD = 163.99) (*p* < 0.05) (Appendix A).

## 4. Discussion

The aim of this study was to describe the physical activity and sedentary activity patterns of pregnant women in a health district in the south of Spain, as well as the variability of these behaviour patterns according to sociodemographic and obstetric variables. The research also aimed to estimate the degree of compliance with the WHO recommendation on physical activity during pregnancy.

The physical activity of the pregnant women was assessed in three different domains: work, travel to and from places, and leisure time physical activity. If we consider the results globally in terms of the number of minutes per week spent on average in each of these three domains, work PA occupied the most time among the pregnant women in the sample (almost 600 min per week on average, out of a total of approximately 900 min of PA per week). In this respect, it should not be forgotten that in the GPAQ questionnaire, work is understood in a broad sense, including domestic activities, job search, and study, beyond strictly work activities in the usual sense of the term. Moreover, the average weekly PA time spent travelling to and from places was close to 100 min (97.6), while the average weekly PA time spent on leisure activities was around 200 min (212.1). Logically, the distribution of the average weekly PA time among these three domains, in addition to personal circumstances, was strongly marked by the social and geographical characteristics of this area of southern Spain and may be very different from that found among pregnant women in other parts of the world. For example, in a study of pregnant women in Nepal, leisure time physical activity was reported to be non-existent (only one woman reported doing it) and practically all the PA performed by pregnant women was distributed between work or domestic chores and travelling to and from places [56].

The distribution by PA domains of participants in terms of prevalence yielded a different perspective to results presented in terms of average weekly minutes dedicated to each area of PA. Specifically, a higher percentage of women in the sample reported performing leisure PA (61.7%), while only a third (33.5%) reported performing PA at work or at home and just over a third reported active travel (36.2%).

In this study, time spent weekly on leisure PA appeared to be closely associated with the level of education (the higher the level of education, the greater the time spent). This agrees with the findings of studies in various countries [23,28,29,35,37,38,39,57,58]. Multiple factors may explain this association between educational level and more frequent leisure PA. For example, a higher level of education facilitates access to health information that supports the desirability of maintaining an active lifestyle at any stage of the life cycle. It is also associated with higher income, which makes it easier to live in areas with greater accessibility to parks, gyms, or other facilities for PA, and safer areas. It may also facilitate the payment of fees for facilities or programmes, when they are not free [35,38,40,57].

In contrast with the previous finding, weekly time of physical activity at work tended to be inversely related to educational level. In this study, the average weekly minutes of work PA were particularly high among women with only compulsory schooling, much higher than among those with higher levels of education. Moreover, PA at work was found to be relatively high among those born outside Spain, among those who spoke a language other than Spanish at home as a child, among the unemployed and among those living in small towns. It is important to note that, in the province of Huelva, in south-western Spain, there are many immigrant women working in the agricultural sector. In addition, as in other geographical areas of the world, women with a lower level of education may be forced to take on more physically demanding work. Yet, regarding unemployed women, the GPAQ questionnaire considers work to include both domestic chores and job search.

Finally, regarding active travel, no relevant sociodemographic variability was detected in this study. In southern European areas, such as the one in the present study, walking or cycling is not necessarily a recreational activity. It can also be a way of going to work, shopping, accompanying children to school, travelling to be with family or friends or accessing health services for pregnant women. It is therefore understandable that no significant social variability in active travel was found, as women from very different social backgrounds may share the habit or need to walk or cycle. In contrast, a study in Ontario, Canada, found that pregnant women with lower educational levels were more likely to walk when travelling [36].

When PA was estimated in terms of energy expenditure equivalent (MET min/week), sociodemographic variability was detected in only two aspects: educational level and size of place of residence. Pregnant women with lower educational levels reported much higher MET min/week (approximately twice as high) than those with higher educational levels, which could be related to a greater involvement in physical work or domestic activities or to a lower use of private cars as a means of transport, among other factors. Regarding size of the place of residence, a higher energy expenditure in physical activity was detected among pregnant women living in small towns. In the geographical area studied, it is precisely in small towns where it is more common for women to work in agriculture or for women to live and travel to larger towns to work as domestic servants. It is also relatively easy to walk to the shops or to the health centre, to go on walks in rural trails, or to use the municipal public sports facilities in these smaller towns. All these activities, among others, may involve higher energy expenditure than is typical among pregnant women living in large cities. However, as the relationship between physical activity in pregnant women and size of place of residence has been scarcely studied, future studies should address this to better understand this relationship.

Regarding differences in the practice of PA in the three domains explored considering obstetric variables, they were only detected in the number of pregnancies of participants and the trimester of pregnancy awareness. In this study, primigravidae reported spending more time per week performing leisure PA than those who had had previous pregnancies. This is consistent with other studies, where pregnant women in their first pregnancy or without children have reported higher levels of physical exercise or leisure physical activity than those with children [23,35,38,59,60,61]. In contrast, in a study conducted in the city of Donostia-San Sebastián, in northern Spain, pregnant women with living children spent more weekly time doing light or moderate PA, measured objectively and with self-reported instruments [25]. Similarly, a study of Hispanic women in the United States found that having children was associated with a higher level of PA, including housework and caregiving.

Both types of results (i.e., higher frequency of PA in primigravidae and higher frequency of PA in multigravidae or multiparous women) could be reasonably explained, although with different arguments. On the one hand, in a social context where social support for parenting is scarce, it seems logical that women who are pregnant for the first time have more free time for recreational PA than those who already have children. On the other hand, if they already have children, this may imply, in addition to more housework, having to leave the house frequently to accompany them (e.g., to school or the park). Yet, in certain social groups, having children may denote a higher level of income than that of childless women, which, in turn, facilitates having free time or being able to pay fees of gyms and the like. Of course, this depends on the geographical area, the type of urban planning in the cities, the social policies of the country, and the socioeconomic characteristics of the population. Hence, the results of studies in different social contexts may be divergent.

Regarding the trimester of pregnancy awareness, very marked differences were detected between women who became aware in the first trimester and those who did in the second trimester. Those who became aware in the second trimester (only five cases—1.3% of the sample) had significantly higher levels of work-related PA as well as energy expenditure (MET-min/week). To correctly interpret these results, it would be necessary to further explore the specific characteristics of these five individuals, according to the anonymous data collected in the questionnaire. However, this was beyond the scope of this study as it was a marginal percentage of pregnant women in the sample.

Regarding compliance with the WHO recommendation to practice at least 150 min a week of moderate PA, the prevalence detected in this study (73.7%) was somewhat lower than that estimated among pregnant women in the city of Donostia-San Sebastián, (85%) [25]. Yet, it was higher than that estimated in the city of Malaga (50.8%) [10], which, like the area studied in this study (the province of Huelva), is in southern Spain. Several studies have found even lower prevalences of compliance with this recommendation among pregnant women. Some examples are a study conducted in Italy (37%) [32], one in South Africa (25%) [34], and one in Brazil (10%) [35], as well as the average prevalence estimated from various studies carried out in China (21%) [31]. However, it should be noted that in our study, as in most of these other studies, the PA of pregnant women was self-reported and was not assessed by pedometers, accelerometers, or other objective measuring techniques [62]. The assessment of PA with questionnaires allows large samples to be studied or to be integrated into clinical practice. However, it may lead to biases associated with social desirability or poor recall, or with transient or stable cognitive impairments that prevent an accurate perception of the subject’s own lifestyle. Interestingly, a Canadian study found that adult pregnant women reported a mean of 34.3 daily minutes of PA, whereas time measured objectively was 14.9 min [33].

Overall, in our study, we found low variability in the prevalence of adherence to the WHO recommendation. Slightly higher compliance was found among participants who were currently studying (86%) compared to those who were not (71.6%). The percentage was somewhat higher among primigravidae (80%) than among those who had previously had more pregnancies (70%). This is consistent with the results of other studies [23,39]. In this study, no relevant variability was detected in the weekly time spent performing PA by pregnant women according to BMI. By contrast, a study in Sweden estimated a higher compliance rate among pregnant women with lower BMI [29].

Regarding time spent daily in sedentary activities, participants reported an average of 3.9 h per day, significantly lower than in other studies. According to a review published in 2017, the average daily time spent by pregnant women in sedentary activities, measured objectively, ranged from 7.07 h in a study in the USA to 18.3 h in a study in Ethiopia. Yet, participants in a study in Singapore reported an average of 8.6 h of sedentary activities per day [63]. More recent studies also show considerable disparity in average daily sedentary activity figures: 5.9 h in a study in Japan [48] and 9.3 h in a study in Sweden [43]. Moreover, a study in Pittsburgh, PA, USA, found that during the COVID-19 years, the PA time of pregnant women decreased, and sedentary activity increased compared to previous years [64].

There has been little research on sociodemographic or obstetric correlates of the time spent in sedentary activities by pregnant women. In our study, marked differences were found according to the educational level of participants (the higher the educational level, the more time spent in sedentary activities per day). This seems to agree with a study in Japan, which showed that time spent in sedentary activities correlated with the income level of the pregnant women [48]. In our study sample, it was also correlated with the language spoken at home as a child (i.e., higher among those who spoke Spanish as a child) and to gravidity (more time spent in sedentary activities per day among primigravidae).

The potential risks of excessive sedentary activity for the pregnant woman, for childbirth, and for child development are highly relevant [42,43,44,45]. Hence, it would be desirable to expand research in this field, addressing the prevailing sedentary activity patterns among pregnant women per se and their predictors and correlates, and to further explore their consequences. This could contribute to improving public health policies and to inform sound professional practice in terms of health advice to pregnant women on physical and sedentary activities. Several studies have confirmed that there tends to be a reduction in physical activity and an increase in sedentary activity during pregnancy [15,46]. Yet, Canadian researchers note that this may now be exacerbated by increasing social pressure for sedentary activities [33].

Moreover, previous studies have shown that there tends to be a continuity between the pre-pregnancy lifestyle and the lifestyle of pregnant women, also in terms of physical activity [22,24,30,38,65]. Women who had an active lifestyle before pregnancy tend to engage in higher levels of PA during pregnancy. This is logical but implies an important challenge in promoting PA among pregnant women, as it highlights the need to promote an active lifestyle among adolescents and women of childbearing age in general. In a global context of increasing physical inactivity among adolescent girls [66], it is key to address the many barriers already identified in many countries that prevent adolescent girls from adopting an active lifestyle [67]. Reducing these barriers involves cross-sectoral action, not only by the health system or from sport and exercise-related agencies, and interdisciplinary collaboration across multiple domains.

Some studies have specifically investigated the barriers encountered by pregnant women when practising PA. One of those most frequently cited is not having received suitable health advice recommending PA or having received contradictory information on the subject [34,35,68]. At the same time, other types of research on the determinants of health professionals’ practice when recommending PA to pregnant women have found that their professional practice in this respect is often influenced by relevant limitations and that they have little support to perform their work in this area [69]. One of the barriers identified in such studies is a lack of undergraduate or postgraduate training. For example, in the United States, only 8% of medical students report having received specific training in physical activity [70]. This may suggest that curricula in medical and other health studies should be reviewed in appropriate areas to allow greater training in the promotion of healthy lifestyles in pregnant women and in the population as a whole [71,72]. The results of these studies also suggest that changes may also be needed in the organisation of health care services. Such changes should enable professionals to provide lifestyle advice to pregnant women with more time in consultations, to have greater institutional support in this role, and to benefit from multidisciplinary collaboration when needed [73].

For obvious reasons, pregnant women with obesity are one of the groups of pregnant women who would require the most support from health professionals in adopting an active lifestyle. Yet, one study suggests that they are the least likely to receive health advice in this regard [74]. This may be linked to the widespread problem of social stigmatisation of obesity, which may also be affecting the health care received by pregnant women living with obesity [75]. Such social stigma has been identified as reducing access to health services for pregnant women with excess weight and leading to mental health problems, with potential counterproductive side-effects on the course of obesity itself [76].

The observed sociodemographic and regional variations in physical activity and sedentary patterns among pregnant women in Southern Spain offer crucial insights for public health policy. These findings underscore the need for antenatal physical activity promotion policies that are not ‘one-size-fits-all’, but rather are **tailored by educational level and geographic region** [77]. Interventions should consider the distinct physical activity profiles of women with different educational backgrounds and those residing in diverse urban and rural settings. For instance, policies could focus on validating and building upon existing occupational and domestic physical activity among lower-educated and rural women, while simultaneously facilitating access to culturally appropriate and enjoyable leisure-time physical activity options. Such targeted strategies are essential to overcome specific barriers and ensure more equitable and effective promotion of healthy physical activity behaviours throughout pregnancy across the population.

The results reveal that the group of pregnant women with the highest energy expenditure in physical activity (women with low educational levels) is precisely the one that engages in the least physical activity during their free time, which might appear paradoxical. However, the collected data also indicate that the energy expenditure of women with low educational levels is largely tied to physical activity at work. It is understandable that pregnant women who perform work requiring significant physical effort may feel less motivated to engage in physical activity during their leisure time afterward. Furthermore, if they live in a social context where beliefs about significantly reducing physical activity during pregnancy prevail, this would be even more comprehensible. On another note, the extent to which occupational physical activity can benefit, or harm foetal development and the progression of pregnancy is still relatively understudied. The findings of a recent cohort study suggest that a high level of occupational physical activity during the first and second trimesters is associated with a lower risk of small for gestational age at birth [78]. While research to date on the impact of PA on foetal development and women’s well-being has predominantly focused on recreational PA, it would be beneficial for future studies to delve deeper into the impact of occupational PA, with a view to formulating sufficiently evidence-based recommendations on this matter.

### Strengths and Limitations

This study has some strengths. The sample selection process was governed by the criterion of obtaining a representative sample of the population of pregnant women who were being attended at their reference public hospital for regular pregnancy monitoring (specifically, at the 20th-week ultrasound scan). An internationally validated questionnaire (i.e., the GPAQ) was used to collect information on different domains of physical activity (at work, in leisure time, and during travel). Data were collected in face-to-face interviews by a trained health professional, in a quiet space in the hospital, using an anonymous questionnaire. Information cards with locally adapted and pictures were used to illustrate different types of moderate or vigorous physical activity, both at work and in leisure time. The omission rate for all questions in the questionnaire was very low (generally less than 1%). Among other aspects, the correlates of sedentary activity in pregnant women, which is a relatively little studied field, were investigated.

However, this study also has some limitations. Its design was cross-sectional, which excludes the possibility of inferring causal relationships between the variables studied. Another limitation was to solely rely on bivariate statistical analyses. While this approach was appropriate for our foundational objective of providing a comprehensive description and identifying initial associations within this understudied population, it inherently limits our ability to establish fully independent relationships between variables or to comprehensively control for potential confounding factors. Future research building upon these preliminary findings should incorporate multivariate analyses to provide a more robust understanding of independent predictors of physical activity and sedentary behaviours in pregnant women. In addition, it was not possible to interview foreign pregnant women who were not fluent in Spanish due to the lack of support from interpreters in the various languages. Although the response rate was relatively high (77.5%), the percentage of women with a university degree among those who agreed to be interviewed was higher than among those who refused to participate. Data were self-reported, as they were collected by means of a questionnaire, which does not exclude different types of bias (mainly social desirability or recall bias). In addition, the GPAQ measures sedentary activities by means of a single item, which could be a limitation of this instrument [79,80].

## 5. Conclusions

The estimated prevalence of compliance with the WHO recommendation for physical activity in pregnancy (i.e., a minimum of 150 min of moderate PA per week) was relatively high among the participants in the sample (approximately 74%). However, this indicates that a quarter of pregnant women have a rather inactive lifestyle, which may lead to considerable problems for themselves and for child development.

There was little sociodemographic variability in compliance with the WHO recommendation. The average weekly time spent on physical activity at work was three times higher than that of leisure time PA in the sample. Work-related PA was performed by approximately one third of the sample, while leisure time PA was practised by approximately two thirds.

The sociodemographic correlates of leisure time PA differed substantially from those of work-related PA, mainly in terms of educational level. Work-related PA was the area of PA with the greatest sociodemographic variability. PA linked to travel did not exhibit any relevant sociodemographic variability.

As for the time spent by pregnant women on sedentary activities, marked differences were detected according to educational level, to the language spoken at home during childhood and to gravidity.

Most of the pregnant women in this study met international PA recommendations for pregnancy. However, the correlates of leisure PA are different from those regarding work PA. Therefore, the planning and implementation of effective public health strategies to foster an active and healthy lifestyle during pregnancy may require different approaches and communication plans, depending on PA literacy and socioeconomic factors.

## Figures and Tables

**Figure 1 healthcare-13-01423-f001:**
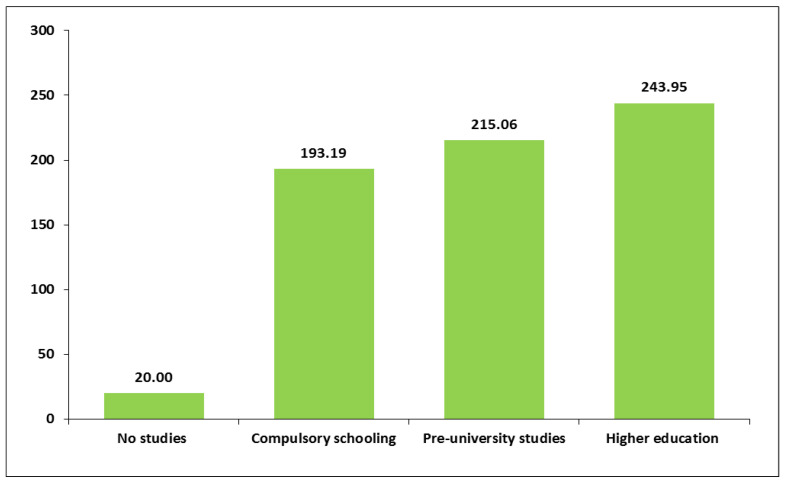
Average number of weekly minutes of physical activity in leisure time according to educational level.

**Figure 2 healthcare-13-01423-f002:**
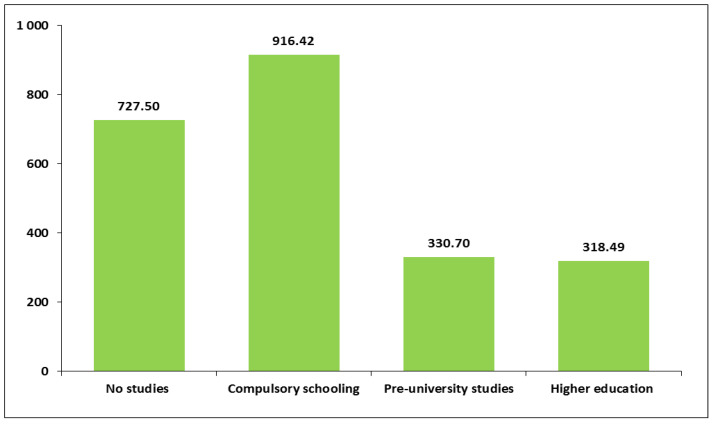
Average weekly minutes of physical activity at work according to educational level.

**Figure 3 healthcare-13-01423-f003:**
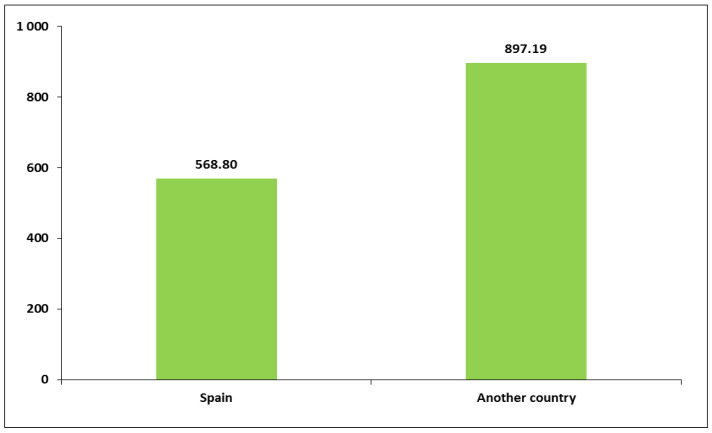
Average weekly minutes of physical activity at work according to country of birth.

**Figure 4 healthcare-13-01423-f004:**
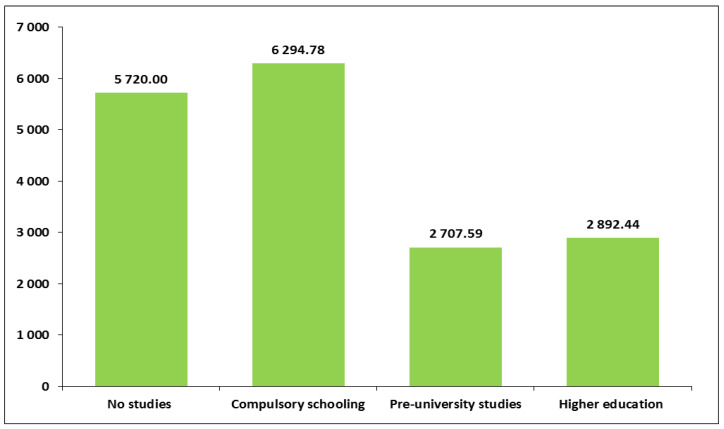
Average MET-minutes per week according to educational level.

**Figure 5 healthcare-13-01423-f005:**
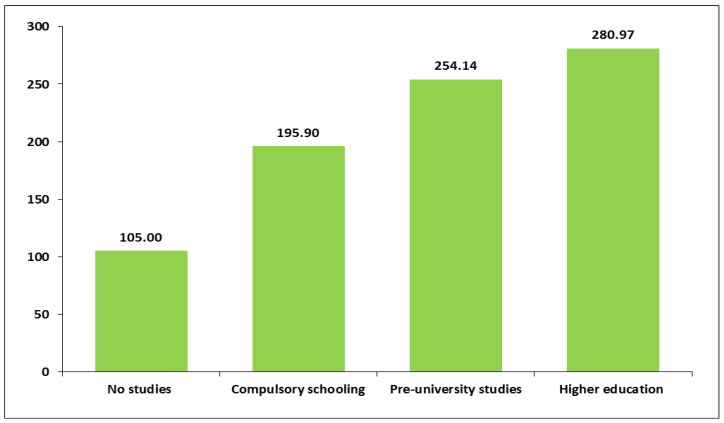
Average daily minutes of sedentary behaviour according to educational level.

**Table 1 healthcare-13-01423-t001:** Sample description of sociodemographic and somatometric characteristics.

Variables	N	Mean	Standard Deviation
Age (years)	385	31.84	5.99
Approximate weight (kg)	383	70.04	14.21
Approximate height (cm)	383	163.30	6.12
Body Mass Index (Kg/height^2^)	382	26.26	5.15
Variables	Categories	N	%
Age	Less than 30 years	119	30.9
	30 to 35 years	110	28.6
	More than 35 years	156	40.5
Educational level	No studies	4	1.0
	Compulsory schooling	175	45.5
	Pre-university studies	87	22.6
	Higher education	119	30.9
Currently studying	No	328	85.2
	Yes	57	14.8
Employment status	Full time	157	41.1
	Part time	82	21.5
	Unemployed	68	17.8
	Other	75	18.6
Size of place of residence	Up to 5000 inhabitants	45	11.7
	From 5001 to 20,000 inhabitants	82	21.3
	From 20,001 to 50,000 inhabitants	137	35.6
	More than 50,000 inhabitants	121	31.4
In a relationship	Yes	382	99.2
	No	3	0.8
Current language spoken at home	Spanish	347	90.1
	Other (including bilingual)	38	9.9
Language spoken at home as a child	Spanish	349	90.6
	Other (including bilingual)	36	9.4
Country of birth	Spain	353	91.7
	Another country	32	8.3
Body Mass Index	Underweight	5	1.3
	Normal weight	182	47.6
	Overweight	116	30.4
	Obesity	79	20.7
Total sample size: 385			

Note: Variables, sample size (N) for each variable, the mean and standard deviation are indicated.

**Table 2 healthcare-13-01423-t002:** Sample description of obstetric characteristics.

Variables	N	Mean	Standard Deviation
Number of pregnancies including the current one	385	2.06	1.11
Number of vaginal births	385	0.48	0.74
Number of caesarean births	385	0.12	0.34
Number of miscarriages	385	0.37	0.72
Number of abortions	385	0.10	0.35
Number of health problems during pregnancies ended in live birth	190	0.25	0.63
Age at first pregnancy (years)	383	27.50	6.35
Date of initiation of health care (in gestational weeks)	380	6.42	2.40
Number of weeks since last menstrual period	382	20.23	0.69
Variables	Categories	N	%
Gravidity (number of pregnancies including the current one)	Primigravida	142	36.9
	Multigravida	243	63.1
Health problems during pregnancies ended in live birth	No	159	83.7
	Yes	31	16.3
Pregnancy planning	No	90	23.4
	Yes	294	76.6
Assisted reproduction pregnancy	No	254	87.3
	Yes	37	12.7
High-risk pregnancy care	No	325	84.4
	Yes	60	15.6
Trimester of pregnancy awareness	First trimester	379	98.7
	Second trimester	5	1.3
Professional health care during pregnancy	No	2	0.5
	Yes	383	99.5

Note: Variables, sample size (N) for each variable, the mean and standard deviation are indicated.

**Table 3 healthcare-13-01423-t003:** Frequency of physical and sedentary activity patterns by domains, METs, and totals.

Variables	N	Mean	Standard Deviation
Total PA (work, travel to and from places, and leisure) (min/week)	384	902.85	1061.31
Total work PA (min/week)	382	596.31	1037.52
Vigorous work PA (min/week)	382	201.28	683.26
Moderate work PA (min/week)	380	397.11	818.45
Total active travel to and from places (min/week)	384	97.57	211.28
Total leisure PA (min/week)	384	212.07	262.34
Vigorous leisure PA (min/week)	384	2.34	34.20
Moderate leisure PA (min/week)	384	209.73	261.99
Total vigorous PA (work and leisure) (min/week)	384	202.58	681.80
Total moderate PA (work, travel to and from places, and leisure) (min/week)	384	700.27	858.64
Total % moderate PA (min/week)	339	90.46	26.66
Number of MET-min/week	384	4421.70	6254.84
Sedentary behaviour (min/day)	383	234.61	162.99
Variables	Categories	N	%
Engaged in PA at work	Yes	128	33.5
	No	254	66.5
Engaged in PA for travel to and from places	Yes	139	36.2
	No	245	63.8
Engaged in PA for leisure	Yes	237	61.7
	No	147	38.3
Engaged in some type of PA	Yes	324	84.4
	No	60	15.6
Engaged in vigorous PA	Yes	42	10.9
	No	342	89.1
Engaged in moderate PA	Yes	312	81.3
	No	72	18.8
Engaged in at least 150 min of moderate intensity PA per week. Met the WHO requirement of PA during pregnancy	Yes	283	73.7
	No	101	26.3

Notes. PA: physical activity; MET: metabolic equivalents of task; Work PA: includes paid and unpaid work such as domestic and caregiving work; Variables sample size (N) for each variable, the mean and standard deviation are indicated.

**Table 4 healthcare-13-01423-t004:** Variability of compliance with the WHO recommendation of at least 150 min of moderate PA per week according to sociodemographic and obstetric characteristics.

Variables/Categories	Yes	No	Statistical Parameters
Age	N = 283 %	N = 101 %	Chi^2^ (2) = 2.56; *p* = 0.28; Cramer’s V = 0.08
Less than 30 years	70.6	29.4	
From 30 to 35 years	70.9	29.1	
More than 35 years	78.1	21.9	
Educational level	N = 283 %	N = 101 %	Chi^2^ (3) = 5.44; *p* = 0.14; Cramer’s V = 0.12
No studies	25.0	75.0	
Compulsory schooling	73.0	27.0	
Pre-university studies	77.0	23.0	
Higher education	73.9	26.1	
Currently studying	N = 283 %	N = 101 %	Chi^2^ (1) = 5.2; *p* = 0.02; Cramer’s V = 0.12
No	71.6	28.4	
Yes	86.0	14.0	
Employment status	N = 280 %	N = 101 %	Chi^2^ (3) = 1.27; *p* = 0.74; Cramer’s V = 0.06
Full time	73.1	26.9	
Part time	78.0	22.0	
Unemployed	70.6	29.4	
Other	72.0	28.0	
Size of place of residence	N = 283 %	N = 101 %	Chi^2^ (3) = 4.52; *p* = 0.21; Cramer’s V = 0.11
Up to 5000 inhabitants	77.8	22.2	
From 5001 to 20,000 inhabitants	81.7	18.3	
From 20,001 to 50,000 inhabitants	70.6	29.4	
More than 50,000 inhabitants	70.2	29.8	
In a relationship	N = 283 %	N = 101 %	Chi^2^ (1) = 0.08; *p* = 0.78; Cramer’s V = 0.01
Yes	73.8	26.2	
No	66.7	33.3	
Current language spoken at home	N = 283 %	N = 101 %	Chi^2^ (1) = 0.15; *p* = 0.7; Cramer’s V = 0.02
Spanish	74.0	26.0	
Other (including bilingual)	71.1	28.9	
Language spoken at home as a child	N = 283 %	N = 101 %	Chi^2^ (1) = 0.96; *p* = 0.33; Cramer’s V = 0.05
Spanish	73.0	27.0	
Other (including bilingual)	80.6	19.4	
Country of birth	N = 283 %	N = 101 %	Chi^2^ (1) = 0.03; *p* = 0.86; Cramer’s V = 0.01
Spain	73.6	26.4	
Another country	75.0	25.0	
Body Mass Index (BMI)	N = 280 %	N = 101 %	Chi^2^ (3) = 1.42; *p* = 0.7; Cramer’s V = 0.06
Underweight	80.0	20.0	
Normal weight	74.2	25.8	
Overweight	69.8	30.2	
Obesity	76.9	23.1	
Gravidity (Number of pregnancies including the current one)	N = 283 %	N = 101 %	Chi^2^ (1) = 5.04; *p* = 0.02; Cramer’s V = 0.11
Primigravida	80.3	19.7	
Multigravida	69.8	30.2	
Health problems during pregnancies ended in live birth	N = 130 %	N = 59 %	Chi^2^ (1) = 0.07; *p* = 0.79; Cramer’s V = 0.02
No	69.2	30.8	
Yes	66.7	33.3	
Pregnancy planning	N = 217 %	N = 73 %	Chi^2^ (1) = 1.36; *p* = 0.24; Cramer’s V = 0.06
No	68.9	31.1	
Yes	75.1	24.9	
Assisted reproduction pregnancy	N = 283 %	N = 101 %	Chi^2^ (1) = 0; *p* = 0.98; Cramer’s V = 0
No	74.8	25.2	
Yes	75.0	25.0	
High-risk pregnancy care	N = 283 %	N = 101 %	Chi^2^ (1) = 0.5; *p* = 0.48; Cramer’s V = 0.04
No	74.4	25.6	
Yes	70.0	30.0	
Trimester of pregnancy awareness	N = 282 %	N = 101 %	Chi^2^ (1) = 0.48; *p* = 0.49; Cramer’s V = 0.04
First trimester	73.8	26.2	
Second trimester	60.0	40.0	
Professional health care during pregnancy	N = 283 %	N = 101 %	Chi^2^ (1) = 0.58; *p* = 0.45; Cramer’s V = 0.04
No	50.0	50.0	
Yes	73.8	26.2	

Note: Variables or categories, sample size (N), and percentage (%) for each variable and for response Yes and No, and statistical parameters are indicated.

## Data Availability

The data presented in this study are available on request from the corresponding author. The data are not publicly available due to privacy restrictions.

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
