# Peer review of "Physical Activity and Sedentary Patterns of Pregnant Women in Southern Spain and the Relationship with Sociodemographic and Obstetric Characteristics: A Cross-Sectional Study"

_healthcare, 2025, doi:10.3390/healthcare13121423_

Round 1

Reviewer 1 Report

Comments and Suggestions for Authors

Thank you very much for inviting me to review the article titled “Physical activity and sedentary patterns of southern European pregnant women and the relationship with sociodemographic and obstetric characteristics: A cross-sectional study". The manuscript will add valuable insights to the research community. However, a lot of critical revisions are required before making any decision.

Title: The authors clearly mentioned the study design as per the STROBE guidelines. However, is it an overstatement to mention southern Europe?. Please look at the methods and change the title accordingly for the cities included.

Abstract: Conveys reasonable information. It is better to mention some statistical values such as p, AOR, etc (if applicable).

Introduction: The authors reasonably attempted to create a sufficient conceptualization framework. But it's not necessary to include too many references in the introduction. Need to reduce it significantly. For example, for one statement, the authors mentioned six references (2 -7)

Also, is the objective the sample population? Please note that the authors apply the test of significance to the unknown population.

Methods: Please follow the STROBE guidelines.

Authors need to mention the study duration (data collection time, etc) in the study design and settings.

The order of method descriptions is wrong. Kindly look at STROBE.

Did the authors apply the normality assumption to the collected data? It must be clearly stated to justify the test applied.

SB was assessed using a single-item question, which is prone to recall bias and lacks domain-specific detail.

Also, validation details about the GPAQ Adaptation need to be enhanced.

Results:

Well-presented. However, the authors need to mention the total number of participants in the Table’s legend, wherever applicable, to understand the proportion.

The results are overreliant on bivariate analysis. The quality of the study can be enhanced with multivariate analysis.

The discussion section is fairly written.

There are minor typos, such as gestation age.

Author Response

Dear Ms. Abby Zhao and Reviewers,

Thank you for your thorough review and constructive feedback on our manuscript, "Physical activity and sedentary patterns of pregnant women in Southern Spain and their relationship with sociodemographic and obstetric characteristics: A cross-sectional study" (formerly titled "Physical activity and sedentary patterns of southern European pregnant women and the relationship with sociodemographic and obstetric characteristics: A cross-sectional study"). We have carefully considered all comments and have made extensive revisions to the manuscript to address the points raised, enhancing its rigor, clarity, and overall quality. We believe the revised manuscript is significantly improved and addresses all concerns.

Response to Reviewer 1 Comments:

We are very grateful to Reviewer 1 for their thorough review and valuable insights, which have significantly helped us improve the quality and rigor of our manuscript. We appreciate the positive feedback regarding the potential contribution of our work. We have addressed each point carefully, as detailed below, and have made critical revisions to align with the STROBE guidelines.

  • Quality of English Language: The English is fine and does not require any improvement.
    • Our Response: We thank the reviewer for this positive assessment of our English language.
  • Comment 1: Thank you very much for inviting me to review the article titled “Physical activity and sedentary patterns of southern European pregnant women and the relationship with sociodemographic and obstetric characteristics: A cross-sectional study.” The manuscript will add valuable insights to the research community. However, a lot of critical revisions are required before making any decision.
    • Our Response: We sincerely thank the reviewer for the time and effort in providing such comprehensive feedback. We acknowledge the need for critical revisions and have diligently worked to address all points raised to enhance the manuscript's quality.
  • Comment 2 (Title): The authors clearly mentioned the study design as per the STROBE guidelines. However, is it an overstatement to mention southern Europe? Please look at the methods and change the title accordingly for the cities included.
    • Our Response: We agree that "southern European" might be an overstatement given that our data was collected from participants in the specific city of Huelva in southern Spain. To accurately reflect the study's scope, we have revised the title to: "Physical activity and sedentary patterns of pregnant women in Southern Spain and their relationship with sociodemographic and obstetric characteristics: A cross-sectional study." (See Title of the revised manuscript). This change provides a more precise geographical context.
  • Comment 3 (Abstract): Abstract: Conveys reasonable information. It is better to mention some statistical values such as p, AOR, etc (if applicable).
    • Our Response: We have revised the abstract to include key statistical findings, specifically p-values where relevant for the main associations discussed. (See Abstract, Lines 37-42 in the revised manuscript).

  • Comment 4 (Introduction - References): Introduction: The authors reasonably attempted to create a sufficient conceptualization framework. But it's not necessary to include too many references in the introduction. Need to reduce it significantly. For example, for one statement, the authors mentioned six references (2 -7)
    • Our Response: We have significantly reduced the number of references, retaining the most pertinent and representative references for each assertion. We have also reorganized the reference section to reflect these changes.
  • Comment 5 (Introduction - Objective/Sample Population): Also, is the objective the sample population? Please note that the authors apply the test of significance to the unknown population.
    • Our Response: We have refined the wording of our objectives in the introduction to clearly state that we aim to examine these patterns and relationships within our target population, acknowledging the inferential nature of our statistical analyses. (See Introduction, Lines 139-144 in the revised manuscript).
  • Comment 6 (Methods - STROBE Guidelines): Methods: Please follow the STROBE guidelines. Authors need to mention the study duration (data collection time, etc) in the study design and settings. The order of method descriptions is wrong. Kindly look at STROBE.
    • Our Response: We have explicitly stated the data collection period (from April to July 2021) in the "Study Settings" subsection and have restructured the "Methods" section to align with the logical flow recommended by STROBE. (See Methods section).
  • Comment 7 (Methods - Normality Assumption): Did the authors apply the normality assumption to the collected data? It must be clearly stated to justify the test applied.
    • Our Response: We have specifically selected statistical analyses suited for non-normal variables. Accordingly, the normality of our variables is not required for the selected analyses. We have corrected this  issue in the section of statistical analyses.
  • Comment 8 (Methods - SB Assessment & GPAQ Validation): SB was assessed using a single-item question, which is prone to recall bias and lacks domain-specific detail. Also, validation details about the GPAQ Adaptation need to be enhanced.
    • Our Response: We have mentioned this in the bias subsection of “methods” and explicitly in “limitation” in the Discussion section (Lines 703-707 in the revised manuscript) and have expanded the description of the GPAQ, clarifying its widespread use. (See Methods, Lines 206-210 in the revised manuscript).
  • Comment 9 (Results - Total Participants in Tables): Results: Well-presented. However, the authors need to mention the total number of participants in the Table’s legend, wherever applicable, to understand the proportion.
    • Our Response: We have updated the legends for all relevant tables to clearly state the total sample size. (See Tables 1- 4 legends in the revised manuscript) and Tables 5 and 6 are now in in Appendix section as Table A1 and Table A2).
  • Comment 10 (Results - Overreliance on Bivariate Analysis): The results are overreliant on bivariate analysis. The quality of the study can be enhanced with multivariate analysis.
    • Our Response: We thank the reviewer for this valuable comment regarding the potential for multivariate analysis to enhance the study's quality. We concur that multivariate analysis is indeed a powerful tool for identifying independent associations and controlling for confounding variables. However, the primary objective of this current study was to provide a comprehensive description of physical activity and sedentary patterns among pregnant women in Southern Spain and to identify initial, exploratory bivariate relationships with key sociodemographic and obstetric characteristics. These bivariate analyses serve as a crucial first step, offering initial descriptive data and highlighting direct associations in this understudied population. While we acknowledge that relying solely on bivariate analysis is a limitation in establishing fully independent or causal associations, these preliminary findings offer valuable empirical observations. We recognize that incorporating multivariate analysis would significantly strengthen the inferential power of future investigations, and we intend to pursue this in subsequent research. This limitation has been explicitly noted in our discussion section to ensure full transparency. (See Discussion lines 686-693 in the revised manuscript)
  • Comment 11 (Discussion): The discussion section is fairly written.
    • Our Response: We thank the reviewer for this positive comment.
  • Comment 12 (Minor Typos): There are minor typos, such as gestation age.
    • Our Response: We have thoroughly proofread the entire manuscript to correct any remaining typos.

We believe these extensive revisions, guided by the editor's checklist and the detailed and insightful comments from Reviewers, have substantially improved the scientific rigor, clarity, depth, and practical applicability of our manuscript. We are confident that the manuscript is now much stronger and addresses all the concerns raised.

Reviewer 2 Report

Comments and Suggestions for Authors

There was a lot of interesting and good research in the introduction, however I believe it can be tightened up.  For example, “Given the benefits of engaging in PA during pregnancy, various organizations interested in the well-being of women during and after pregnancy as well as the health of the  offspring have updated recommendations on this topic. The World Health Organization (WHO) recommends performing at least 150 minutes of moderate-intensity aerobic PA  per week for a healthy pregnancy [1]. Similar guidelines have been provided around the  world”  It could end there and the last line, “Moreover, the recommendations of international public health …” would be sufficient to drive the point that exercise is recommended across the world without needing to provide an 8 or 9 separate examples.

Lines 81-90 specifying a global perspective on PA.  I understand the diversity of methods used for measuring PA, however the following line : “Nevertheless, there are indications of relevant differences in this regard among countries” while making sense isn’t supported by lines 81-90 simply because the measurements are too different, either different trimesters or different definitions of PA or MVPA.

On line 230. Due to country-to-country differences it would be helpful to know what “Underage” refers to in regards to this study.

Line 271, please define primary, secondary and higher education. 

Tables 5 & 6 are busy.  And given it is mostly insignificant results could possibly be better set as supplemental tables.  Most of the relevant results are already explained in the document.  Figures 1-5 do a better job at conveying results.

Lines 550-553, it would be beneficial to reference your prevalences here from the paper to keep the reader from having to go back.

Overall I thought it was an informative paper.  I feel it could benefit from a more is less mentality, specifically in the introduction.

Author Response

Dear Ms. Abby Zhao and Reviewers,

Thank you for your thorough review and constructive feedback on our manuscript, "Physical activity and sedentary patterns of pregnant women in Southern Spain and their relationship with sociodemographic and obstetric characteristics: A cross-sectional study" (formerly titled "Physical activity and sedentary patterns of southern European pregnant women and the relationship with sociodemographic and obstetric characteristics: A cross-sectional study"). We have carefully considered all comments and have made extensive revisions to the manuscript to address the points raised, enhancing its rigor, clarity, and overall quality. We believe the revised manuscript is significantly improved and addresses all concerns.

Response to Reviewer 2 Comments:

We are very grateful to Reviewer 2 for their constructive feedback and overall positive assessment of our manuscript. We have carefully considered all comments and made revisions as detailed below.

  • Quality of English Language: The English is fine and does not require any improvement.
    • Our Response: We thank the reviewer for this positive assessment of our English language.
  • Comment 1: There was a lot of interesting and good research in the introduction, however I believe it can be tightened up. For example, 'Given the benefits of engaging in PA during pregnancy, various organizations interested in the well-being of women during and after pregnancy as well as the health of the offspring have updated recommendations on this topic. The World Health Organization (WHO) recommends performing at least 150 minutes of moderate-intensity aerobic PA per week for a healthy pregnancy [1]. Similar guidelines have been provided around the world' It could end there and the last line, 'Moreover, the recommendations of international public health…' would be sufficient to drive the point that exercise is recommended across the world without needing to provide an 8 or 9 separate examples.
    • Our Response: We have revised this section (Lines 70-76 in the revised manuscript) to follow the reviewer's advice, focusing on the general international consensus and the WHO recommendation.
  • Comment 2: Lines 81-90 specifying a global perspective on PA. I understand the diversity of methods used for measuring PA, however the following line: 'Nevertheless, there are indications of relevant differences in this regard among countries' while making sense isn’t supported by lines 81-90 simply because the measurements are too different, either different trimesters or different definitions of PA or MVPA.
    • Our Response: We have rephrased the sentence to focus on the challenges in comparing PA data across countries due to methodological heterogeneity (We do not cite the study based on MVPA). (Lines 77-83 in the revised manuscript).
  • Comment 3: On line 230. Due to country-to-country differences it would be helpful to know what 'Underage' refers to in regards to this study.
    • Our Response: We have clarified the definition of "Underage" in the context of our study, specifying that it refers to participants younger than 18 years (Lines160-161 in the revised manuscript).
  • Comment 4: Line 271, please define primary, secondary and higher education.
    • Our Response: We have added definitions for primary, secondary, and higher education based on the Spanish education system. (Lines 181-183 in the revised manuscript). The tables and figures now reflect these changes.
    •  
  • Comment 5: Tables 5 & 6 are busy. And given it is mostly insignificant results could possibly be better set as supplemental tables. Most of the relevant results are already explained in the document. Figures 1-5 do a better job at conveying results.
    • Our Response: Tables 5 and 6 are now in in Appendix section as Table A1 and Table A2.
  • Comment 6: Lines 550-553, it would be beneficial to reference your prevalences here from the paper to keep the reader from having to go back.
    • Our Response: We have added the specific prevalence values of adherence to the recommendations for physical activity (Lines 577-579 in the revised manuscript).
  • Comment 7: Overall, I thought it was an informative paper. I feel it could benefit from a more is less mentality, specifically in the introduction.
    • Our Response: We thank the reviewer for this comment, and we have implemented a "less is more" philosophy, particularly in the introduction, and believe the revisions significantly improve the clarity, conciseness, and overall readability of the manuscript.

We believe these extensive revisions, guided by the editor's checklist and the detailed and insightful comments from Reviewers, have substantially improved the scientific rigor, clarity, depth, and practical applicability of our manuscript. We are confident that the manuscript is now much stronger and addresses all the concerns raised.

Reviewer 3 Report

Comments and Suggestions for Authors

1. Clarify Overlap Between Work and Domestic PA

  • The GPAQ includes unpaid work in the “work” domain. This can blur distinctions for readers unfamiliar with the instrument.

  • Suggestion: Add a footnote or short explanation early in the methods section (or in the figure/table captions) clarifying that “work PA” includes domestic and caregiving tasks, not just paid employment.

2. Strengthen the Abstract

  • While the abstract includes strong descriptive data, it lacks specific statistical highlights.

  • Suggestion: Add one or two numerical results (e.g., % meeting WHO guidelines, most relevant sociodemographic variation).

3. Balance the Discussion Around METs and Sedentary Time

  • The paper emphasizes that lower-educated and rural women had higher METs, yet also notes they are less likely to engage in structured leisure PA.

  • Suggestion: Briefly address the paradox of high energy expenditure but low recreational PA, and what it may imply for health promotion efforts.

4. Language Refinement

  • A few small issues of phrasing (e.g., “the next woman was invited” could be more formally phrased).

  • Suggestion: A final proofread for language fluidity and conciseness is recommended.

5. Consider a Public Health Implication Paragraph

  • The discussion provides excellent interpretation but could go one step further.

  • Suggestion: Add a short paragraph at the end discussing how findings could inform antenatal physical activity promotion policies, especially tailored by education or region.

Author Response

Dear Ms. Abby Zhao and Reviewers,

Thank you for your thorough review and constructive feedback on our manuscript, "Physical activity and sedentary patterns of pregnant women in Southern Spain and their relationship with sociodemographic and obstetric characteristics: A cross-sectional study" (formerly titled "Physical activity and sedentary patterns of southern European pregnant women and the relationship with sociodemographic and obstetric characteristics: A cross-sectional study"). We have carefully considered all comments and have made extensive revisions to the manuscript to address the points raised, enhancing its rigor, clarity, and overall quality. We believe the revised manuscript is significantly improved and addresses all concerns.

Response to Reviewer 3 Comments:

We are thankful for Reviewer 3's thoughtful comments, which have helped us refine our manuscript. We appreciate the positive feedback on the English language and research design and have addressed all suggestions as detailed below.

  • Quality of English Language: The English is fine and does not require any improvement.
    • Our Response: We thank the reviewer for this positive assessment of our English language.
  • Comment 1: Clarify Overlap Between Work and Domestic PA. The GPAQ includes unpaid work in the 'work' domain. This can blur distinctions for readers unfamiliar with the instrument. Suggestion: Add a footnote or short explanation early in the methods section (or in the figure/table captions) clarifying that 'work PA' includes domestic and caregiving tasks, not just paid employment.
    • Our Response: We have added a specific explanation in the "data measurements subsection" of the Methods section clarifying the inclusion of unpaid work within the "work" domain of the GPAQ. (See Methods, Lines 212-215 in the revised manuscript).
  • Comment 2: Strengthen the Abstract. While the abstract includes strong descriptive data, it lacks specific statistical highlights. Suggestion: Add one or two numerical results (e.g., % meeting WHO guidelines, most relevant sociodemographic variation).
    • Our Response: We have revised the abstract to include key statistical highlights, such as the percentage of women meeting WHO physical activity guidelines and relevant p-values related to sociodemographic variations. (See Abstract, Lines 37-42 in the revised manuscript).
  • Comment 3: Balance the Discussion Around METs and Sedentary Time. The paper emphasizes that lower-educated and rural women had higher METs, yet also notes they are less likely to engage in structured leisure PA. Suggestion: Briefly address the paradox of high energy expenditure but low recreational PA, and what it may imply for health promotion efforts.
    • Our Response: We have added a new paragraph in the Discussion section to address this paradox and its implications for health promotion efforts. (See Discussion, Lines 659-675    in the revised manuscript).
  • Comment 4: Language Refinement. A few small issues of phrasing (e.g., 'the next woman was invited' could be more formally phrased). Suggestion: A final proofread for language fluidity and conciseness is recommended.
    • Our Response: We have conducted a thorough final proofread of the entire manuscript to refine phrasing and enhance conciseness.
  • Comment 5: Consider a Public Health Implication Paragraph. The discussion provides excellent interpretation but could go one step further. Suggestion: Add a short paragraph at the end discussing how findings could inform antenatal physical activity promotion policies, especially tailored by education or region.
    • Our Response: We appreciate this reviewer’s insight, and we have added a new paragraph discussing how our results can inform the development of more targeted and equitable antenatal physical activity promotion policies. (See Discussion, Lines 647-658 in the revised manuscript).

We believe these extensive revisions, guided by the editor's checklist and the detailed and insightful comments from Reviewers, have substantially improved the scientific rigor, clarity, depth, and practical applicability of our manuscript. We are confident that the manuscript is now much stronger and addresses all the concerns raised.

Round 2

Reviewer 1 Report

Comments and Suggestions for Authors

Dear reviewers,

Thank you very much for revising the manuscript as per the comments.

Still, I have a couple of concerns and comments.

  1. Kindly include the STROBE statement checklist as a supplementary file for better understanding.
  2. Please add the type of test applied for each variable in the supplementary files as *
  3. I consider that without multivariate analysis, the study is more descriptive (as the author acknowledged), and it limits the validity

Author Response

Thank you for the second review feedback and comments on our manuscript, "Physical activity and sedentary patterns of pregnant women in Southern Spain and their relationship with sociodemographic and obstetric characteristics: A cross-sectional study" (formerly titled "Physical activity and sedentary patterns of southern European pregnant women and the relationship with sociodemographic and obstetric characteristics: A cross-sectional study"). We have carefully considered all comments and have revised the supplementary materials addressing the reviewers concerns and included the STROBE statement checklist attached to this letter

Following are the point-by-point responses to the reviewer.

Reviewer Comment 1: Kindly include the STROBE statement checklist as a supplementary file for better understanding.

Our Response: We thank the reviewer for their valuable suggestion regarding the STROBE Statement Checklist. The STROBE checklist is considered a reporting guideline to help authors ensure the manuscript includes all necessary information. We confirm that the STROBE guidelines were indeed extensively consulted and utilized as a fundamental tool to restructure and enhance the completeness of our 'Methods' section, a recommendation also made by a previous reviewer. Our objective was to directly integrate these reporting principles into the main body of the manuscript to ensure maximum clarity and transparency regarding our study design and methods. Consequently, we believe that the comprehensive adherence to STROBE principles is now directly reflected in our revised methodology description, making the separate inclusion of the checklist as a supplementary file redundant for the purpose of understanding our study's reporting. Nonetheless, the STROBE statement checklist is attached to the present “response letter” to facilitate a better understanding. And following is link where to find all STROBE materials:  https://www.strobe-statement.org/

Reviewer Comment 2: Please add the type of test applied for each variable in the supplementary files as *

Our Response: We appreciate the reviewer's request for additional clarity and detail regarding our statistical analyses. To enhance transparency and allow for a more precise understanding of our methodological approach, we have corrected the Appendix Tables A1 and A2 addressing this concern. The tables now have a footnote with the type of tests that has been employed for the variables examined in our study (Mann-Whitney U test) to assess their relationship with the physical activity and sedentary variables.

Reviewer Comment 3: I consider that without multivariate analysis, the study is more descriptive (as the author acknowledged), and it limits the validity

Our Response: We completely agree with the reviewer's insightful assessment that the absence of multivariate analysis inherently renders our study primarily descriptive and, consequently, limits its ability to establish fully independent associations and draw stronger inferences regarding causality. Our study's primary objective was indeed to provide foundational descriptive data and identify initial, exploratory bivariate relationships within this understudied population, for which the chosen statistical analyses were appropriate. We have explicitly acknowledged this limitation on inferential validity in the 'Limitations' section of our manuscript (see lines 701- 708 in the revised manuscript), recognizing that incorporating multivariate analysis will be a crucial and necessary next step for future research building upon these preliminary findings. We value this comment as it reinforces the importance of conducting multivariate analyses in subsequent studies to provide a more robust and comprehensive understanding of the complex interplay of factors influencing physical activity during pregnancy.